

# Risk factors for falls in community-dwelling older people with mild cognitive impairment: a prospective one-year study

Thanwarat Chantanachai[1,2], Morag E. Taylor[1,3], Stephen R. Lord[1,4], Jasmine Menant[1,4], Kim Delbaere[1,4], Perminder S. Sachdev[5,6], Nicole A. Kochan[6], Henry Brodaty[6] and Daina L. Sturnieks[1,2]

[1] Falls, Balance and Injury Research Centre, Neuroscience Research Australia, Sydney, New South Wales, Australia
[2] School of Medical Sciences, University of New South Wales, Sydney, New South Wales, Australia
[3] Prince of Wales Clinical School, University of New South Wales, Sydney, New South Wales, Australia
[4] School of Population Health, University of New South Wales, Sydney, New South Wales, Australia
[5] Neuropsychiatric Institute, Prince of Wales Hospital, Randwick, New South Wales, Australia
[6] Faculty of Medicine and Health, University of New South Wales, Centre for Healthy Brain Ageing (CHeBA), Discipline of Psychiatry and Mental Health, Sydney, New South Wales, Australia

Corresponding author
Daina L. Sturnieks,
d.sturnieks@neura.edu.au

## ABSTRACT

**Objective:** Mild cognitive impairment (MCI) is considered an intermediate stage between normal cognitive function and dementia. Fall risk is increased in this group, but there is limited literature exploring specific fall risk factors that may be addressed in fall prevention strategies. The aim of this study was to examine risk factors for falls in older people with MCI, focusing on cognitive, psychological and physical factors.
**Methods:** Participants ($n$ = 266, 45% women) were community-dwelling older people aged 70–90 years who met the criteria for MCI. Cognitive, psychological, sensorimotor and physical assessments, physical activity levels, medication use, general health and disability were ascertained at baseline. Falls were monitored prospectively for 12 months.
**Results:** During follow-up, 106 (40%) participants reported one or more falls. Poorer visual contrast sensitivity, increased postural sway, lower levels of weekly walking activity, higher levels of depressive symptoms and psychotropic medication use were significantly associated with faller status (≥1 falls) in univariable analyses. Of these factors, poor visual contrast sensitivity, increased postural sway and psychotropic medication use were found to be significant independent predictors of falls in multivariable analysis while controlling for age and sex. No measures of cognitive function were associated with falls.
**Conclusions:** Poor visual contrast sensitivity, impaired balance and psychotropic medication use predicted falls in community-dwelling people with MCI. These risk factors may be amenable to intervention, so these factors could be carefully considered in fall prevention programs for this population.

## INTRODUCTION

Falls are a major issue affecting both physical and mental health in older people (*Terroso et al., 2014*). The consequences of falls are serious and associated with mortality, morbidity, hospitalisation, and substantial economic costs (*James et al., 2020*). One third of cognitively healthy community-dwelling older people fall one or more times annually, and this rate doubles in people with dementia (*Allan et al., 2009*; *Kalache et al., 2007*; *Tinetti, Speechley & Ginter, 1988*; *Wesson et al., 2013*). The proportion of falls are also high in people in the early stage of cognitive impairment, ranging from 53% to 63% in older people with mild cognitive impairment (MCI) over 6 to 12 month follow-up periods (*Ansai et al., 2019*; *Ansai et al., 2018*; *Davis et al., 2017*; *Goncalves et al., 2018*).

There is now considerable knowledge of fall risk factors in cognitively healthy older people (*Lord, Sherrington & Naganathan, 2021*) and some studies have elucidated fall risk in older people with manifest cognitive impairment and dementia (*Chantanachai et al., 2021*; *Dolatabadi et al., 2018*; *Fernando et al., 2017*; *Modarresi et al., 2019*). However, few studies have examined fall risk in older people with MCI, a growing proportion of the older population with measurable memory loss (amnestic) and/or impairments in executive function, attention, language, and visuospatial skills (non-amnestic) that is in excess of normal age-related declines, but not sufficiently severe to affect the ability to carry out daily tasks (*Petersen, 2004*; *Roberts & Knopman, 2013*). While these changes do not affect activities of daily living, evidence suggests that subtle changes may occur well before diagnosis and understanding these early impairments will allow for timely intervention.

It has been reported that older people with MCI present with physical and cognitive deficits such as poor balance, gait, executive function and memory (*Bahureksa et al., 2017*; *Johnson et al., 2012*). There is some evidence from prospective studies that these factors contribute to fall risk in people with MCI including reduced balance (*Makizako et al., 2013*), impaired mobility (*Ansai et al., 2018*), slow choice stepping reaction time (*Bunce et al., 2017*), slow gait speed (*Pieruccini-Faria et al., 2020*), and low levels of physical activity and functional status (*Ansai et al., 2019*). Reduced ability to dual-task (*Ansai et al., 2018*) has also been found to increase fall risk, yet impaired functioning in cognitive domains such as executive function, visuospatial abilities, attention, memory, fluency and language has not in people with MCI (*Ansai et al., 2019*). However, these studies have been limited by small sample sizes (*Ansai et al., 2019*; *Ansai et al., 2018*; *Makizako et al., 2013*), inadequate follow-up length and/or procedures (*Makizako et al., 2013*), unstandardised fall ascertainment (*Makizako et al., 2013*) or have had only a narrow focus in terms of risk factors (*Bunce et al., 2017*; *Pieruccini-Faria et al., 2020*). In addition, poor visual contrast sensitivity has been reported to be associated with falls in cognitively healthy older people, but has not been examined in older people with MCI (*Saftari & Kwon, 2018*). Consequently, our understanding of fall risk in people with MCI is limited and insufficient to inform interventions specifically addressing the needs of this group.

To address this gap, we conducted a large prospective cohort study in community-dwelling older people with MCI to identify key explanatory and modifiable risk factors for falls from a comprehensive range of cognitive, mood, physical, and functional mobility measures. We hypothesised that fall risk factors in older people with MCI are multifaceted and include balance impairment, executive dysfunction and depressive symptoms based on previous work (*Ansai et al., 2019*; *Delbaere et al., 2012*; *Taylor et al., 2014*). By elucidating why older people with MCI fall, our findings may assist in the development of appropriately targeted fall prevention strategies.

## MATERIALS AND METHODS

### Participants

Participants were enrolled in the Sydney Memory and Ageing Study (MAS) and consented to participate in a sub-study involving balance and fall risk assessment (*Tsang et al., 2013*). Inclusion criteria were aged 70–90 years and living in the community. Exclusion criteria were: diagnosed dementia as determined by Diagnostic and Statistical Manual of Mental Disorders (DSM-IV) criteria and a consensus diagnosis from an expert team comprised of old age psychiatrists, neuropsychiatrists and neuropsychologists; being unable to speak and understand English; or having a current diagnosis of progressive malignancy (active cancer or current treatment for cancer, other non-metastasised prostate cancer and skin cancer), motor neuron disease, multiple sclerosis, developmental disability, major psychiatric disorders, psychotic symptoms and medical or psychological conditions that prevented participants from completing assessments (*Sachdev et al., 2010*). MCI classification was determined by: (a) cognitive impairment as shown by performance 1.5 standard deviations (or equivalent) below published normative values (matched for age and education where available) on a neuropsychological test measure (*Petersen, 2004*), (b) normal or minimally functional impairments as determined by informant ratings on the Bayer-ADL scale (score of <3) adjusted for physical impairment (*Hindmarch et al., 1998*), (c) consensus diagnosis for MCI was undertaken for those with Bayer-ADL scale scores ≥3, and (d) no diagnosis of dementia (DSM-IV) as per the consensus of an expert team comprised of old age psychiatrists, neuropsychiatrists and neuropsychologists. Participants who were classified as having MCI with one-year falls follow-up from waves 1 or 2 were included. Participants provided written informed consent, and the University of New South Wales Human Studies Ethics Committee approved the study (Ref. number: HC17865 and HC200671).

## ASSESSMENTS

### Demographic and health conditions

Participants were interviewed by trained research staff to complete questionnaires to document age, sex, education, history of falls, medication use, and presence of medical conditions such as stroke, heart disease, diabetes mellitus, and arthritis. Psychotropic medication use was categorized as not taking/taking one or more of the following classes of medications: sedatives/hypnotics, anxiolytics, antipsychotics, and antidepressants. Each of
these classes is also reported as not taking/taking one or more medications. Height and weight were measured to compute body mass index (BMI).

## Cognitive function

Global cognitive function was assessed with the Mini-Mental State Examination (MMSE). The MMSE score was prorated in cases where one or two items were missing at random and incorporates adjustments for demographic variables (age, education and non-English speaking background status) which have been shown to influence MMSE performance (*Anderson et al., 2007*). Memory was assessed using the Logical Memory Story A Test. Participants were read a short story and had to recall everything that they remembered, immediately, and after a 30-min delay (*Wechsler, 1997*). Processing speed and executive function were measured with the Trail Making Test (TMT). This test consists of two parts, TMT-A assesses processing speed and TMT-B assesses executive function (*Bowie & Harvey, 2006*; *Tombaugh, 2004*). The difference score (TMT-B minus TMT-A) was calculated to remove the speed component from the test evaluation to provide a more precise measure of executive function (*Strauss, Sherman & Spreen, 2006*). Attention and processing speed were assessed using the Digit Symbol-Coding test (*Wechsler, 1997*). Verbal fluency was assessed using the Controlled Oral Word Association Test, which requires participants to say as many words as possible within a 60-s time limit, beginning with three letters (F, A, S). The Controlled Oral Word Association Test score is the sum of the responses for the letters, excluding repetitions and non-words and is a measure of executive function (*Strauss, Sherman & Spreen, 2006*).

## Psychological assessment

Depressive symptoms were assessed using the short form (15-item) Geriatric Depression Scale (GDS); higher scores indicate greater symptom severity (*Yesavage & Sheikh, 1986*). Anxiety symptoms were assessed using the Goldberg Anxiety Scale (GAS). GAS scores range from 0 to 9 with higher scores representing greater anxiety (*Goldberg et al., 1988*). Concern about falling was assessed using the 16-item Falls Efficacy Scale International (FESI). Participants rated their concern about falling for each item on a 4-point scale (1 = not at all concerned to 4 = very concerned). The total score ranges from 16 to 64 with higher scores indicating greater concern (*Yardley et al., 2005*).

## Sensorimotor performance

Sensorimotor performance was assessed using the Physiological Profile Assessment (PPA) which consists of five sensorimotor assessments: visual contrast sensitivity, lower limb proprioception, lower limb muscle strength, hand reaction time, and postural sway, as follows (*Lord, Menz & Tiedemann, 2003*). The Melbourne Edge Test of visual contrast sensitivity requires participants to discriminate between the high and low contrast halves of 20 circular images with reducing contrast (dB) by identifying the orientation of the contrast edge. Lower limb proprioception was assessed with a vertical clear acrylic sheet inscribed with a protractor placed between participants legs while in a seated position. Participants were instructed to close their eyes and match the position of their greater toes

on either side of an acrylic protractor with any offset in position measured in degrees. Lower limb muscle strength was measured using a spring gauge while participants were seated with the hip and knee joints positioned at 90°. Participants were asked to extend their knee as hard as they could against the resistance of the spring gauge attached to their lower leg. The highest maximal isometric knee extension force (kg) of three trials was recorded. Hand reaction time was measured using a light stimulus and finger-press response. Following five practice trials, the average of 10 repeated trials was recorded in milliseconds. Postural sway (balance) was tested using a sway meter that records body displacement at the level of the waist. Participants stood on a foam mat with eyes open for 30 s, during which sway path (mm) was recorded. Weighted contributions from these five PPA assessments were used to calculate the PPA fall risk score, with higher scores indicating poorer physical performance and greater risk of falling (*Lord, Menz & Tiedemann, 2003*).

## Functional mobility and balance

Functional mobility and balance were assessed using tests of sit-to-stand (STS), timed up and go (TUG) and coordinated stability. The STS test involved five repeated sit to stand movements from a standard height (43 cm) chair with arms held across the chest (*Whitney et al., 2005*). Participants were instructed to complete the task as fast as possible and their performance was timed (s) (*Lord et al., 2002*; *Netz, Axelrad & Argov, 2007*). For the TUG, participants were asked to rise from a chair, walk three metres at usual pace, turn around, walk back to the chair, and sit down. Participants were instructed to complete the TUG as quickly and safely as possible and time was recorded in seconds (*Podsiadlo & Richardson, 1991*). The coordinated stability test measures participants' ability to adjust their body position in a steady and coordinated way when near or at the limits of their base of support (*Lord, Ward & Williams, 1996*). A sway meter was attached to the participants' waist with the rod extending anteriorly. Participants were asked to adjust the position of their body without moving their feet so that the swaymeter pen tip followed and remained within a convoluted track marked on the piece of paper placed at the level of the waist. A total error score was calculated by summing the number of occasions that the pen failed to stay within the path (one error point) and the number of track corners cut across (five error points). Participants attempted the test twice, with the better score taken as the test result (*Lord, Ward & Williams, 1996*).

## Physical activity

Physical activity was assessed using the Incidental and Planned Exercise Questionnaire (*Delbaere, Hauer & Lord, 2010*). This questionnaire provides estimates of planned activity, planned and incidental walking activity and total activity in hours/week over the past 3 months.

## Health-related disability

The 12-item World Health Organisation Disability Assessment Schedule II (WHODAS II) assessed health and disability (*Epping-Jordan & Ustun, 2000*). Participants rated how much
difficulty (1 = none to 5 = extreme/cannot do) they experienced while undertaking activities including self-care, mobility, cognitive tasks, maintaining relationships, and community participation, over the past 30 days. Scores range from 12 (no disability) to 60 (complete disability).

## Ascertainment of falls

Falls were monitored prospectively for 12 months using monthly falls diaries with reply paid envelopes. When fall diaries were not returned within 2 weeks of the end of each month, participants were contacted by phone to obtain the required information. A fall was defined as "an unexpected event in which the person comes to rest on the ground, floor, or lower level" (*Lamb et al., 2005*). Fallers were defined as people who had at least one fall during the 12-month follow-up period.

## Statistical analyses

Statistical analyses were completed using SPSS statistics version 26 (IBM Corp, Armonk, NY, USA). Significance was set at $p < 0.05$. Data are reported as mean ± standard deviation (SD) for continuous data and $n$ (%) for categorical data. Between-group (faller and non-faller) comparisons of baseline demographic characteristics were made using independent samples $t$-tests for continuous variables and chi-square for categorical variables. Scores of >3 SD above/below the mean were allocated to participants who were physically unable to perform physical tests to assign appropriate scores for these data that were not missing at random, *i.e.* mean plus/minus 3SD scores are indicative of poor performances but not excessively high to have undue influence in the statistical analysis (*Kwak & Kim, 2017*). To not disadvantage the ~1% of participants who could complete the tests with scores > mean plus/minus 3SD, scores for these participants were also assigned mean plus/minus 3SD scores. The number of participants for whom such imputations were made for the relevant tests were: five repetition STS $n = 28$; TUG $n = 6$; coordinated stability $n = 6$; proprioception $n = 5$; hand reaction time $n = 9$; knee extension strength $n = 3$; postural sway $n = 2$). Fall risk factors were examined using univariable logistic regression models with faller status as the outcome variable (no falls *vs* one or more falls) and cognitive, psychological and physiological factors as predictor variables.

The significant risk factors identified in univariable analyses were entered into a multivariable logistic regression model (using the enter method) to identify significant and independent predictors of falls (variables providing unique explanatory information in predicting fall risk). Continuous variables that were significant in the univariable logistic regression analyses were dichotomised to allow better comparison of the odds of increasing falls of the included variables in the multivariable model. We used a previously established cut-point for GDS (GDS 15-item: ≥4) (*Taylor et al., 2014*) or the median as the cut-point (visual contrast sensitivity, postural sway and walking activity). First, these dichotomised variables were examined univariably to establish that they remained significantly associated with falls, then they were entered into the multivariable model while adjusting for age and sex. Classification accuracy (≥1 falls) is reported for the final multivariable model.

## RESULTS

Two hundred and sixty-six of 763 who completed the 12-month follow-up for falls met the criteria for MCI and were included in this study. During follow-up, 106 (40%) MCI participants reported at least one fall (≥1 falls) and 47 (18%) reported multiple falls (≥2 falls). Table 1 presents the demographic and medical characteristics of the fallers (≥1 falls) and non-fallers (no falls). A greater proportion of the prospective faller group reported a fall in the past year and taking psychotropic medications, compared to the non-faller group (Table 1).

### Fall risk factors: univariable logistic regression

Table 2 presents the results of the separate univariable logistic regression analyses examining cognitive, psychological, sensorimotor, functional performance, physical activity and WHODAS measures for their association with faller status. Depressive symptoms, poor visual contrast sensitivity, impaired balance (increased postural sway), higher PPA Fall Risk Scores and lower levels of walking activity were significantly associated with falling (≥1 falls).

### Independent predictors of falls: multivariable logistic regression

Table 3 presents the results from the multivariable logistic regression analysis that included the five significant predictors from univariable analyses. Poor visual contrast sensitivity, increased postural sway, and psychotropic medication use were identified as significant independent predictors of falls while controlling for age and sex with a classification accuracy of 69.7%. Compared to the null model, there was a significant improvement in fit for the prediction of faller status (−2log likelihood model = 283.17). In a sensitivity analysis that precluded psychoactive medication use from model entry, depressive symptoms were identified as an independent and significant predictor of faller status (OR = 2.0, 95% CI [1.01–3.40], $p$ = 0.046).

## DISCUSSION

In our sample of older people with MCI living in the community, 40% reported at least one fall during a 12-month follow-up period. Higher GDS (depression) scores, poorer vision (lower contrast sensitivity), impaired balance (increased postural sway), lower levels of walking activity, and psychotropic medication use were significantly associated with falls, and from these variables, reduced vision, poorer balance and psychotropic medication use were found to be independent predictors of falls in a multivariable analysis.

The prevalence of falls reported here is consistent with one longitudinal study of older people with MCI that had a maximum follow-up period of 84-months (*Pieruccini-Faria et al., 2020*). However, other studies have documented higher falls incidences. In a study of 361 older people with MCI, *Davis et al. (2017)* found 60% fell one or more times during 12-month follow-up, and 38% fell multiple times in this period. Similarly, in a sample of 38 people with MCI, 53% fell at least once during a 6-month follow-up period (*Ansai et al., 2019*; *Ansai et al., 2018*; *Goncalves et al., 2018*). The discrepancy in fall rates is likely due to differing health and socioeconomic profiles of the study cohorts, *e.g.* greater mobility
**Table 1 Baseline characteristics for participants prospectively categorised as non-fallers (no falls) and fallers (one or more falls) (n = 266).**

| Characteristic, n (%) or mean ± SD | Total | Non-fallers (0) n = 160 | Fallers (≥1) n = 106 | p-value* |
|---|---|---|---|---|
| **Demographics** | | | | |
| Age (years) | 266 | 78.9 ± 4.3 | 78.8 ± 4.8 | 0.880 |
| Sex (female) | 266 | 75 (47) | 45 (43) | 0.478 |
| BMI (kg/m$^2$) | 256 | 27.4 ± 4.5 | 26.6 ± 4.1 | 0.140 |
| Education (years) | 266 | 11.2 ± 3.4 | 12.1 ± 3.9 | 0.061 |
| History of falls (≥1 falls) | 262 | 21 (13) | 73 (70) | **0.001** |
| Walking aid use | 237 | 13 (9) | 14 (15) | 0.169 |
| **Medical history** | | | | |
| Stroke | 264 | 5 (3) | 2 (2) | 0.539 |
| Transient Ischemic Attack | 266 | 15 (9) | 6 (6) | 0.271 |
| Heart problem | 264 | 44 (28) | 30 (28) | 0.936 |
| Hypertension | 266 | 98 (61) | 64 (60) | 0.886 |
| Diabetes | 265 | 15 (9) | 15 (14) | 0.217 |
| Increased cholesterol | 265 | 84 (53) | 48 (45) | 0.229 |
| Arthritis | 266 | 79 (49) | 63 (59) | 0.107 |
| Osteoporosis | 255 | 28 (18) | 19 (19) | 0.899 |
| Parkinson's disease | 265 | 1 (1) | 1 (1) | 0.763 |
| Depression | 249 | 15 (10) | 15 (15) | 0.241 |
| **Medication use** | | | | |
| Total number | 262 | 5.1 ± 3.1 | 5.9 ± 3.6 | 0.076 |
| Psychotropic medication use (one or more) | 250 | 19 (13) | 27 (27) | **0.004** |
| Sedative or hypnotic use | 250 | 10 (7) | 13 (13) | 0.090 |
| Antianxiety agent use | 250 | 4 (3) | 8 (8) | 0.070[†] |
| Antipsychotic use | 250 | 0 (0) | 2 (2) | 0.159[†] |
| Antidepressant use | 250 | 6 (4) | 13 (13) | **0.009** |

Notes:
BMI, body mass index; kg, kilograms; m, metres; n, number of participants; SD, standard deviation. Medical history; self-reported conditions as diagnosed by a doctor. Bold p-values highlight significant findings (p < 0.05). Psychotropic medication use (reported as not taking/taking one or more) included the following medication classes: sedative/hypnotic, antianxiety, antipsychotic, and antidepressant medications.
* Chi$^2$ for categorical variables and independent samples t-test for continuous variables.
[†] Fisher's exact test was applied when more than 20% of the expected counts were less than 5.

impairment, higher proportions of females and participants with a history of falls, older age, lower levels of education, and higher rates of psychotropic medication use in other studies (*Ansai et al., 2019*; *Ansai et al., 2018*; *Davis et al., 2017*; *Goncalves et al., 2018*).

Increased postural sway has been identified as a risk factor for falls in cognitively healthy older people (*Chantanachai et al., 2021*; *Johansson et al., 2017*; *Kwok, Clark & Pua, 2015*; *Muir et al., 2010*), and in a recent systematic review in older people with cognitive impairment (*Chantanachai et al., 2021*). The current study complements these findings and builds on previous retrospective studies conducted in older people with MCI (*Liu-Ambrose et al., 2008*) by confirming an association between impaired balance and prospective falls using gold standard follow-up procedures. Future research could examine the contribution of directional sway which may elucidate contrasting balance strategies between older people with and without MCI.

**Table 2 Univariable logistic regression analyses, examining cognitive, psychological, sensorimotor, functional mobility, physical activity and health/disability measures as predictors of prospective falls (0 vs ≥1).**

| Characteristic, n (%) or mean ± SD | Total | Non-fallers (0) $n = 160$ | Fallers (≥1) $n = 106$ | OR (95% CI) | p-value |
|---|---|---|---|---|---|
| **Cognitive function** | | | | | |
| MMSE score | 265 | 28.2 (1.5) | 28.3 (1.5) | 1.06 [0.90–1.25] | 0.511 |
| TMT-A, s | 260 | 48.6 (16.8) | 49.5 (17.6) | 1.00 [0.99–1.02] | 0.675 |
| TMT-B, s | 258 | 143.2 (70.1) | 144.5 (57.1) | 1.00 [1.00–1.00] | 0.871 |
| TMT-B minus TMT-A, s | 258 | 94.6 (63.7) | 94.7 (52.4) | 1.00 [0.99–1.00] | 0.985 |
| FAS total responses | 265 | 33.5 (12.1) | 33.6 (11.1) | 1.00 [0.98–1.02] | 0.932 |
| Digit symbol total correct | 260 | 44.4 (11.1) | 45.0 (10.3) | 1.01 [0.98–1.03] | 0.648 |
| Logical memory immediate recall score | 266 | 9.5 (3.8) | 9.7 (4.1) | 1.01 [0.95–1.08] | 0.654 |
| Logical memory delayed recall score | 266 | 7.7 (3.9) | 7.7 (3.9) | 1.01 [0.95–1.07] | 0.845 |
| **Psychological assessment** | | | | | |
| GAS score | 259 | 1.1 (2.0) | 1.2 (2.1) | 1.03 [0.91–1.17] | 0.626 |
| GDS score | 263 | 1.9 (1.9) | 2.5 (2.0) | 1.19 [1.04–1.36] | **0.012** |
| FESI score | 266 | 22.7 (6.8) | 24.1 (7.6) | 1.03 [0.99–1.06] | 0.132 |
| **Sensorimotor performance** | | | | | |
| Visual contrast sensitivity (dB) | 266 | 20.7 (2.3) | 20.1 (2.1) | 0.89 [0.79–1.00] | **0.045** |
| Proprioception (degrees) | 266 | 2.6 (1.6) | 2.5 (1.5) | 0.98 [0.83–1.14] | 0.748 |
| Hand reaction time (ms) | 266 | 241.2 (45.7) | 244.0 (50.4) | 1.06 [0.83–1.35][a] | 0.640 |
| Knee extension strength (kg) | 265 | 27.9 (11.9) | 25.9 (10.4) | 0.99 [0.96–1.01] | 0.169 |
| Postural Sway (mm) | 266 | 190 (93) | 216 (94) | 1.31 [1.02–1.68][a] | **0.033** |
| PPA Fall Risk Score | 265 | 0.9 (1.0) | 1.2 (0.9) | 1.31 [1.02–1.69] | **0.037** |
| **Functional mobility and balance** | | | | | |
| Five Sit-to-Stand (s) | 266 | 18.6 (7.5) | 18.2 (7.4) | 1.00 [0.99–1.00] | 0.368 |
| Timed Up and Go (s) | 257 | 10.2 (3.2) | 10.3 (3.6) | 1.00 [0.98–1.01] | 0.582 |
| Coordinated Stability (errors) | 263 | 16.0 (13.5) | 17.6 (13.8) | 1.01 [0.99–1.03] | 0.331 |
| **Physical activity** | | | | | |
| Total (hours/wk) | 241 | 32.5 (16.9) | 28.8 (16.0) | 0.99 [0.97–1.00] | 0.097 |
| Walking activity (hours/wk) | 258 | 3.5 (4.6) | 2.3 (3.8) | 0.93 [0.88–1.00] | **0.035** |
| Planned (hours/wk) | 248 | 3.7 (4.5) | 2.6 (4.0) | 0.94 [0.88–1.01] | 0.070 |
| **Health and disability** | | | | | |
| WHODAS score | 258 | 18.6 (6.4) | 19.1 (6.8) | 1.01 [0.98–1.05] | 0.507 |

Notes:

SD, standard deviation; n, number of participants; OR, odds ratio; CI, confidence interval; MMSE, Mini-Mental State Examination; TMT-A, Trail Making Test A; TMT-B, Trail Making Test B; FAS, Controlled Oral Word Association Test responses to words beginning with F, A and S. GAS, Goldberg Anxiety Scale; GDS, Geriatric Depression Scale; FESI, Falls Efficacy Scale-International; dB, decibel; ms, milliseconds; kg, kilograms; mm, millimetres; PPA, Physiological Profile Assessment; s, second; wk, week; WHODAS, The World Health Organisation Disability Assessment Schedule. Bold p-values highlight significant findings ($p < 0.05$). Higher scores represent better performance for MMSE, FAS total responses, Digit symbol total correct, Logical memory immediate recall score, Logical memory delayed recall score, Melbourne Edge Test, knee extension strength, and physical activity. Lower scores represent better performance for TMT-A time score, TMT-B time score, TMT-B minus TMT-A time score, GAS, GDS, FESI, proprioception, hand reaction time, postural sway, PPA score, Sit-to-Stand time score, Timed-up-and-go, and coordinated stability.

[a] OR calculated using z scores due to small unit of measurement for these variables.

Our finding that reduced vision was associated with falls in older people with MCI is also in accordance with previous research conducted in cognitively healthy older people (*Hong et al., 2014*; *Lord & Dayhew, 2001*; *Lord et al., 1994*). Poor visual contrast sensitivity may contribute to falls *via* not only reducing the ability to detect environmental hazards but also by impairing balance, as seen in older people with poor visual contrast

**Table 3 Univariable and multivariable predictors of falls in older people with MCI.**

| Fall risk factor | Univariable | | Multivariable[a] | |
|---|---|---|---|---|
| | OR (95% CI) | *p*-value | OR (95% CI) | *p*-value |
| Depressive symptoms: GDS ≥ 4 | 2.24 [1.19–4.23] | **0.013** | 1.49 [0.72–3.13] | 0.283 |
| Poor visual contrast sensitivity: MET ≤ 21 | 2.56 [1.36–4.79] | **0.003** | 3.67 [1.73–7.78] | **0.001** |
| Greater postural sway: ≥182 mm | 1.77 [1.08–2.90] | **0.025** | 1.92 [1.07–3.44] | **0.028** |
| Lower levels of walking activity/week: <0.45 h | 1.80 [1.09–2.99] | **0.022** | 1.67 [0.94–2.98] | 0.081 |
| Psychotropic medication use | 2.55 [1.33–4.90] | **0.005** | 3.72 [1.67–8.24] | **0.001** |
| Age | – | – | 0.95 [0.89–1.01] | 0.096 |
| Male sex | – | – | 1.39 [0.78–2.47] | 0.266 |

Notes:
OR, odds ratio; CI, confidence interval; GDS, Geriatric Depression Scale; MET, Melbourne Edge Test; mm, millimetres; h, hour. Bold *p*-values highlight significant findings (*p* < 0.05).
[a] Model adjusted for age and sex with each of the fall risk factors identified as having a significant association with faller status in univariable analysis.

sensitivity and stereopsis (*Lord, Clark & Webster, 1991*), and by restricting activity due to a fear of falling (*Wang et al., 2012*). Interestingly, this finding contrasts with studies conducted in older people with moderate cognitive impairment, where reduced visual contrast sensitivity was not associated with falls (*Taylor et al., 2014*; *Taylor et al., 2017*; *Taylor et al., 2012*). It is possible people with MCI have higher levels of physical activity and greater exposure to outdoor hazards that could contribute to trips and slips. Future research could explore the relationship between vision and falls in early cognitive decline by examining visual function more thoroughly (including visual acuity, visual contrast sensitivity, depth perception, visual field loss and motion perception) as well as measuring the type of spectacles worn at the time of falls and the locations where falls occur.

Our finding of a significant association between psychotropic medication use and falls builds on the few previous studies that have reported associations between the use of psychotropic mediations and falls or fall-related injury in older people with mild to moderate cognitive impairment and Alzheimer's Disease (*Hart et al., 2019*; *Horikawa et al., 2005*; *Taylor et al., 2014*), and the studies conducted in cognitively healthy older people (*Seppala et al., 2018*). In this study, fallers were twice as likely as non-fallers to take psychotropic medications, which are known to lead to central nervous system suppression and interfere with sensorimotor functions (*e.g.* balance and reaction time) (*Seppala et al., 2018*). Reducing psychotropic medication use and/or using non-pharmacological treatments to manage anxiety, sleep problems and depression might help to reduce falls in this population. However, future research is required to identify the best strategies for sustaining reduced psychotropic medication use (*Campbell et al., 1999*).

A few factors that were not included in the final multivariable model require discussion. First, the finding that depressive symptoms was a significant univariate predictor of falls is consistent with the findings of a previous study conducted in older people with MCI (*Ansai et al., 2019*), and studies conducted in older people with mild to moderate cognitive impairment and dementia (*Allan et al., 2009*; *Taylor et al., 2014*). Psychotropic medications are often prescribed to treat depressive symptoms and it is likely that these

variables did not provide sufficiently discrete explanatory information regarding fall risk (*i.e.* had some shared variance). This is supported by the inclusion of depressive symptoms in a sensitivity analysis that excluded psychotropic medications as a possible predictor and suggests depressive symptoms, as well as psychotropic medication use warrants attention when considering strategies for fall prevention in older people with MCI. Second, in line with previous studies reporting low activity levels to be a fall risk factor, reduced walking activity was associated with falls in univariable analysis. The maintenance of physical activity, including walking, may benefit older people with MCI, as multiple studies have documented the beneficial effects of physical activity for cardiovascular, neurological and psychological health (*Pinckard, Baskin & Stanford, 2019*; *Taylor, 2014*). However, it should be noted that walking alone has not been shown to prevent falls (*Sherrington et al., 2020*).

Finally, we did not find significant associations between the measured cognitive functions and falls, including global cognitive function, executive function, memory, processing speed, and verbal fluency. Future research could further examine the relationship between specific features of MCI sub-types (*e.g.* amnestic MCI and non-amnestic MCI) and falls, as one previous study suggested that only non-amnestic MCI is a risk factor for falls (*Delbaere et al., 2012*).

The strengths of this study include the relatively large sample of people with MCI and range of quantitative assessments across cognitive, psychological and physical domains as well as health and disability, combined with multivariable modelling that helps identify key factors for falls in this population. However, the results should be considered in view of certain limitations. First, participants were relatively healthy, well-educated and lived in one urban area from one city, which may reduce generalisability of the findings. Second, there is a possibility that not all falls were recorded during follow up. The participants, due to their MCI, may have been less accurate in reporting falls, especially when a fall did not involve injuries, despite the use of gold standard fall ascertainment methods (*e.g.* monthly calendars and telephone calls) (*Lamb et al., 2005*). Third, this study did not examine risk factors for falls within subgroups of people with MCI (*e.g.* amnestic MCI and non-amnestic MCI), due to insufficient subgroup sample sizes.

## CONCLUSIONS

This study has shown that more depressive symptoms (higher GDS scores), poorer vision (reduced contrast sensitivity) and balance (increased postural sway), lower levels of walking activity, and psychotropic medication use are associated with falls in older people with MCI. Reduced vision, poorer balance, and psychotropic medication use were identified as independent predictors of falls among older people with MCI. As these factors are potentially modifiable, the efficacy of balance training exercises, strategies to avoid psychotropic medication use and correction of vision should be considered in interventions for preventing falls in people with MCI.

## ACKNOWLEDGEMENTS

We thank the participants and their informants for their time and generosity in contributing to this research. We also acknowledge the Sydney Memory and Ageing Study Research Team.

### Funding

The Sydney Memory and Ageing Study has been funded by three National Health & Medical Research Council (NHMRC) Program Grants (ID Nos. ID350833, ID568969, and APP1093083). This manuscript does not reflect the views of the NHMRC or funding partners. Morag Taylor is partially funded by the NHMRC Centre for Research Excellence in the Prevention of Fall-related Injuries. Kim Delbaere is supported by an NHMRC Investigator grant. Stephen Lord is a NHMRC Senior Principal Research Fellow. The funders had no role in study design, data collection and analysis, decision to publish, or preparation of the manuscript.

### Grant Disclosures

The following grant information was disclosed by the authors:
National Health & Medical Research Council (NHMRC) Program: ID350833, ID568969, and APP1093083.
NHMRC Centre for Research Excellence in the Prevention of Fall-related Injuries.
NHMRC Investigator grant.

### Competing Interests

The Physiological Profile Assessment (PPA, marketed as NeuRA FallScreen) is commercially available through Neuroscience Research Australia (NeuRA). Any profits from sales of the PPA are shared equally between the inventor (Lord), the Falls and Balance Research Group at NeuRA, and the NeuRA central fund. Stephen R. Lord, Morag E. Taylor, Jasmine Menant and Kim Delbaere are employed by NeuRA. Daina L. Sturnieks holds a conjoint appointment at NeuRA. NeuRA is an independent, not-for-profit research institute. Perminder Singh Sachdev is a member of an Advisory Committee for Biogen Australia. Henry Brodaty has been a consultant or advisory board member for Biogen, Nutricia Australia, Roche and Skin2Neuron.

### Author Contributions

- Thanwarat Chantanachai conceived and designed the experiments, performed the experiments, analyzed the data, prepared figures and/or tables, authored or reviewed drafts of the paper, and approved the final draft.
- Morag E. Taylor conceived and designed the experiments, performed the experiments, analyzed the data, prepared figures and/or tables, authored or reviewed drafts of the paper, and approved the final draft.

- Stephen R. Lord conceived and designed the experiments, performed the experiments, analyzed the data, prepared figures and/or tables, authored or reviewed drafts of the paper, and approved the final draft.
- Jasmine Menant performed the experiments, authored or reviewed drafts of the paper, and approved the final draft.
- Kim Delbaere performed the experiments, authored or reviewed drafts of the paper, and approved the final draft.
- Perminder S. Sachdev performed the experiments, authored or reviewed drafts of the paper, and approved the final draft.
- Nicole A. Kochan performed the experiments, authored or reviewed drafts of the paper, and approved the final draft.
- Henry Brodaty performed the experiments, authored or reviewed drafts of the paper, and approved the final draft.
- Daina L. Sturnieks conceived and designed the experiments, performed the experiments, analyzed the data, prepared figures and/or tables, authored or reviewed drafts of the paper, and approved the final draft.

## Data Availability

The data were first collected in 2007 and participants did not consent to data sharing at that time so the deidentified participant-level data cannot be published with the article.

To request the MAS data, please email the CHeBA Research Bank for a current application form: CHeBAData@unsw.edu.au.

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
