# Peer review of "Risk factors for falls in community-dwelling older people with mild cognitive impairment: a prospective one-year study"

_PeerJ, doi:10.7717/peerj.13484_

## Round 0.1 · original submission · Major Revisions

Dear Dr. Chantanachai,

Thank you for your submission to PeerJ.

It is my opinion as the Academic Editor for your article - Risk factors for falls in community-dwelling older people with mild cognitive impairment: A prospective one-year study - that it requires a number of Major Revisions.

Reviewer 1 ·

Basic reporting

In the abstract, the sentence “..no cognitive factors associated with falls..” needs to be clarified. Perhaps it would be best to say “ no measures of cognitive function”. In the introduction the term “ do not affect daily life function” needs to be clarified. Perhaps to activities of daily living, if this is accurate. There is a notation of “enter method” noted on line 228 of the statistical analysis section. Perhaps this was left over after a review or edit. In Table 1, what medications are included in the Psychotropic medication category? If this includes the other medications listed then then numbers don’t add up. Medication use should be describe further in the methods. What were the psychotropic medications that predicted fall? Antidepressants?
What are the psychotropic agents here? Antidepressants, antianxiety, stimulants, antipsychotic agents and mood stabilizers, benzodiazepines, barbituates, etc.

Experimental design

Unclear why participants who walk with a walking aid were excluded here. This should be explained. Why adjust for only age and sex ? Why not also include specific medication use and educational level for example. Medication use should be describe further in the methods

Validity of the findings

No comment

Additional comments

The authors need to clarify which medications were in the category of psychotropic medication use. It is not clear if this included sedative hypnotics, antianxiety, antipsychotic agents and antidepressants. If this is the case, the numbers don’t add up to a total of 19 non-faller group and 27 in the faller group. Should adjustments have been made for antidepressents whose given that numbers was significantly higher in the faller group. Is this what was driving the observed association between MCI and falls. Use of antipsychotic agents is associated with cognitive impairment. Were these agents excluded under the psychotropic agent category. Patients on psychotropic agents may screen positive for MCI due to medication use.

Annotated reviews are not available for download in order to protect the identity of reviewers who chose to remain anonymous.

·

Basic reporting

1. Early review in the aspect of vision (visual acuity and visual contrast sensitivity) could be useful as compared to introducing in Discussion section. Perhaps consider a paragraph before Page 8 Line 91.

2. Page 8 Line 94: Balance or postural sway could be introduced to complement the various factors of falls.

3. Consider using “poor visual contrast sensitivity” instead of poor vision.

4. Abstract Conclusion: “… so require consideration in fall prevention programs for this population.” The authors can consider “… so these factors could be carefully considered in fall prevention programs for this population.”

Experimental design

5. Page 14 Line 221: It is unclear how the 3 SD allocation was determined.

6. Page 14 Line 231: It is unclear to me why median is used as a cut-off for the Melbourne Edge Test.

Validity of the findings

7. My main concern is the interpretation of findings, specifically Table 2, “Postural sway”. The odds ratio is 1, or approximates very closely to 1, which can be statistically significant but not clinically meaningful. Thus, potentially inappropriate to proceed with dichotomisation of the continuous data.
Huak, C. Y. (2009). Are you a p-value worshipper?. European Journal of Dentistry, 3(03), 161-164.

8. Understandably, a study cited by the authors (Taylor et al. 2017) has already identified postural sway as a modifiable risk factor of future falls. The use of sway area may not consider directional sway (anteroposterior and mediolateral) differences of individuals who may utilise different sway strategy. There are other studies that the authors could consider, which could also enhance the discussion in the authors’ postural sway discussion.
Johansson, J., Nordström, A., Gustafson, Y., Westling, G., & Nordström, P. (2017). Increased postural sway during quiet stance as a risk factor for prospective falls in community-dwelling elderly individuals. Age and ageing, 46(6), 964-970.
Kwok, B. C., Clark, R. A., & Pua, Y. H. (2015). Novel use of the Wii Balance Board to prospectively predict falls in community-dwelling older adults. Clinical biomechanics, 30(5), 481-484.
Maki, B. E., Holliday, P. J., & Topper, A. K. (1994). A prospective study of postural balance and risk of falling in an ambulatory and independent elderly population. Journal of gerontology, 49(2), M72-M84.
Mignardot, J. B., Beauchet, O., Annweiler, C., Cornu, C., & Deschamps, T. (2014). Postural sway, falls, and cognitive status: a cross-sectional study among older adults. Journal of Alzheimer's Disease, 41(2), 431-439.

Additional comments

The author presented a good study aim and methods to address the research gaps. I have provided some suggestions and hope that they could help to enhance your team's manuscript.

---

## Round 0.2 · accepted · Accept

Dear authors, you have addressed the reviewers' comments correctly. The manuscript is much improved.

·

Basic reporting

no comment

Experimental design

no comment

Validity of the findings

no comment

Additional comments

The authors made considerable efforts to improve the manuscript and have sufficiently addressed my comments. I have no further comments.